# A Study of Hepatitis A Seroprevalence in a Paediatric and Adolescent Population of the Province of Florence (Italy) in the Period 2017–2018 Confirms Tuscany a Low Endemic Area

**DOI:** 10.3390/vaccines9101194

**Published:** 2021-10-17

**Authors:** Beatrice Zanella, Sara Boccalini, Massimiliano Alberto Biamonte, Duccio Giorgetti, Marco Menicacci, Benedetta Bonito, Alessandra Ninci, Emilia Tiscione, Francesco Puggelli, Giovanna Mereu, Paolo Bonanni, Angela Bechini

**Affiliations:** 1Department of Health Sciences, University of Florence, 50134 Florence, Italy; beatrice.zanella@unifi.it (B.Z.); sara.boccalini@unifi.it (S.B.); benedetta.bonito@unifi.it (B.B.); emilia.tiscione@unifi.it (E.T.); paolo.bonanni@unifi.it (P.B.); 2Medical Specialization School of Hygiene and Preventive Medicine, University of Florence, 50134 Florence, Italy; massimilianoalberto.biamonte@unifi.it (M.A.B.); duccio.giorgetti@unifi.it (D.G.); marco.menicacci@unifi.it (M.M.); alessandra.ninci@unifi.it (A.N.); 3Meyer Children’s Hospital, 50139 Florence, Italy; francesco.puggelli@meyer.it; 4AUSL Toscana Centro, 50122 Florence, Italy; giovanna.mereu@uslcentro.toscana.it

**Keywords:** hepatitis A, vaccination, seroprevalence, Italy, Florence, travelers, children, VFR

## Abstract

*Background*: Italy is considered an area with very low HAV (hepatitis A virus) endemicity. Currently in Italy the anti-HAV vaccine is recommended only for specific risk groups and there is no universal vaccination program. The aim of this study was to assess the level of immunity against hepatitis A in a sample of children and adolescents from the province of Florence. *Methods*: A total of 165 sera were collected from subjects aged 1 to 18 years, proportionally selected according to the general population size and stratified by age and sex. A qualitative evaluation of anti-HAV antibodies was performed using the enzyme-linked immunosorbent assay (ELISA). Anamnestic and vaccination status data were also collected. *Results*: Our study showed a hepatitis A seroprevalence of 9.1% in the enrolled population. A statistically significant difference in the prevalence of anti-HAV was found between Italian and non-Italian subjects. About half of the population having anti-HAV antibodies was reported to be vaccinated, and no cases of hepatitis A were found. *Conclusions*: The data from our study confirmed Tuscany as an area with low HAV endemicity and showed that hepatitis A seroprevalence is significantly higher in foreign children and adolescents. The presence of more seropositive subjects than those vaccinated was probably due to a natural immunization achieved through a subclinical infection and/or to underreporting of the surveillance systems.

## 1. Introduction

Hepatitis A is a serious infectious disease caused by the hepatitis A virus (HAV), which infects the hepatocytes. HAV is an RNA virus and is generally transmitted through the fecal-oral route due to contamination of food and water. Nevertheless, transmission can also occur in countries where the risk of infection from food or water is low through outbreaks among men who have sex with men (MSM) and persons who inject drugs (PWIDs). The incubation period is generally 15–50 days and the infection may occur asymptomatically; whereas when symptoms and clinical signs are shown, these may include fever, malaise, loss of appetite, diarrhoea, nausea, abdominal discomfort, jaundice, dark urine, and light stools. Unlike the hepatitis B and C viruses, HAV does not cause a chronic infection.

The post-infection and post-vaccination immunity are assessed through the detection of total anti-HAV (IgG) antibodies [1]. Worldwide, hepatitis A occurs sporadically and in epidemics, with a tendency for cyclic recurrences. Globally, 1.4 million hepatitis A cases are diagnosed each year, and the World Health Organization (WHO) estimates that it caused approximately 7100 deaths in 2016 (accounting for 0.5% of the mortality due to viral hepatitis) [2].

Hepatitis A vaccine has been available in Europe since 1991, and it is indicated in the prophylaxis (active immunization) of hepatitis A. WHO recommends the universal vaccination in intermediate endemicity countries; whereas in low and very low hepatitis A endemicity countries, vaccination is recommended for risk groups, such as people living in places with local epidemics, people suffering from other diseases (i.e., chronic liver disease), relatives of infected subjects, travelers going to high endemic areas, healthcare workers, and men who have sex with men (MSM) [3]. 

The first-ever safe and highly effective hepatitis A vaccine in Italy became available in 1995. After a large epidemic of hepatitis A in 1998, the Puglia region (southeastern Italy) has introduced a free-of-charge mass vaccination program for newborns (15–18 months of age) and adolescents (12 years of age), as part of the routine immunisation schedule. Since 2001, the regional health authorities have implemented the vaccination plans according to their own epidemiological context [4,5]. Therefore, the current National Immunization Plan (NIP) 2017–2019, recommends the hepatitis A vaccination to high risk groups, i.e., travelers to endemic areas, drug users, MSM, armed forces, sewage workers, patients with liver disease, liver transplanted subjects, and HAV-negative haemophiliacs [6]⁠. The Tuscany Region follows the national guidelines; thus hepatitis A is recommended in Tuscany for close contacts of clinical cases as a control measure in case of an epidemic [7].

According to the European Centre for Disease Prevention and Control (ECDC) Annual Epidemiological Report, 13,038 confirmed cases of Hepatitis A were reported in 2016 by 29 European Union (EU) and European Economic Area (EEA) countries, corresponding to a notification rate of 2.40 cases per 100,000 individuals. The notification rate of hepatitis A varied greatly across the region with the highest rates observed in eastern EU countries. [8]. In Italy, data on symptomatic acute hepatitis A cases are monitored by the Integrated Epidemiological System of Acute Viral Hepatitis (SEIEVA). The analysis of the trend indicates an uncertain trend in cases of disease ranging from a few (19, an all-time low in 2012) to many (346, an all-time high in 2017). The year 2017 was indeed characterized by an epidemic outbreak that affected a large part of Europe, including Italy. Epidemics had previously occurred in Tuscany in 2008, mainly due to risky behaviors (promiscuity amongst MSM) and later in 2013 for the consumption of contaminated frozen berries. In Tuscany, the annual rates calculated on the cases of the years 2013–2017 show a greater incidence in subjects aged 25–44 years old. The median age of cases in the analyzed period varies from 25 years old in 1994 to 36 years old in 2017; therefore, it reveals a slight but constant increase. In 2019, the number of reported cases of Hepatitis A decreased compared to the previous year, with an incidence of 0.8 cases per 100,000 inhabitants (the incidence was 1.5 cases per 100,000 inhabitants in 2018) [9].

The aim of our study was first to assess the immunity/susceptibility to hepatitis A in a representative sample of the paediatric and adolescent population resident in the province of Florence, by analyzing the sera collected at the blood sampling center of the Meyer Children’s Hospital in Florence. Vaccination status or previous disease notifications were also investigated. 

A second purpose of the study was to compare the results obtained with the data of some previous seroepidemiological surveys carried out in similar geographical areas. 

## 2. Materials and Methods

This hepatitis A seroprevalence analysis is part of a wider seroepidemiological project carried out by the Department of Health Sciences of the University of Florence, started in 2017. This study included the seroprevalence investigation of other vaccine-preventable disease such as Hepatitis B [10], Measles [11], Rubella [12] and Varicella [13]. Blood samples collection took place at the blood sampling center of Meyer Children’s Hospital (Florence, Italy) in the period December 2017–April 2018. The study was conducted in accordance with the Declaration of Helsinki and the protocol was approved by the local Ethics Committee (Project identification code: DSS-UNIFI 98/2017). The sampled population included residents in the province of Florence (Italy) aged 1–18 years and it was proportionally selected according to the general population size and stratified by age and sex. Blood samples were collected from 165 subjects who represented roughly 0.1% of the total resident population in the considered range of age (166,644 subjects in 2017 in the same age group) [14]. No further standardization was performed. Parents or guardians of the enrolled subjects provided a written consent for the blood samples collection before the enrolment; moreover, an information form was given to each subject aged 7 years or older. We excluded non-residents in the province of Florence, immunocompromised patients, subjects under immunosuppressive treatment, those with an acute hepatitis A infection history in the previous two weeks, and those who had received a blood transfusion within the six months prior to the study. Participants were asked to fill in a questionnaire from which some epidemiological indicators were obtained, such as the (self-reported) history of hepatitis A disease and HAV vaccination.

The hepatitis A vaccination status of each subject was confirmed by the consultation of immunization registries available in Tuscany: the Informatic System for Collective Prevention (SISPC; Consortium Metis, Tuscany, Italy) and CARIBEL (Aster, Tuscany, Italy). The confirmation of the previous natural infection or disease was obtained using infectious diseases surveillance records (SIMI, software: Epi Info, Rome, Italy) with the support of Epi-Info application program [15]. 

We also collected information about the number of HAV vaccine doses administered, the date of the last received dose and, if known, the type of vaccine for each subject, along with age, sex, and birth date.

All the collected blood samples were centrifuged (1600 rpm at 4 °C), and the recovered sera were stored at −20 °C until tested for hepatitis A antibodies. Serological analysis was performed by applying the immunoenzymatic ELISA test, ETI-AB-HAVK PLUS (DiaSorin, Italy), for a qualitative assessment of total anti-HAV antibodies in the collected sera. Absorbance readings were performed with Tecan’s Infinite F50 microplate reader, using Magellan for F50 software.

The cut-off value was determined as the mean absorbance of the calibrator at 20 mIU/mL. The optical density (O.D.) of the individual wells reflect the positivity or negativity for anti-HAV antibodies, based on comparison with the cut-off value. A grey area is also considered, defined by an absorbance range ± 20% of the threshold value. 

A sample was considered positive (containing anti-HAV antibodies) for lower values of O.D. than the grey zone; in contrast, when the O.D. was higher than the grey zone, the sample was defined negative (absence of anti-HAV antibodies). Samples having an O.D. included in the grey zone were retested. The repeated reactivity in at least one of the replicated tests defines the sample as positive, whereas in the case of non-reactivity in the second test the sample must be considered negative.

All data were collected in an Excel database and analyzed based on the following parameters: anti-HAV antibodies presence/absence (positive and negative), sex, nationality (Italian and non-Italian), age group, vaccination status, number of doses received, and time elapsed since the last dose. For the purposes of the analysis, all subjects holding an Italian nationality were classified as Italians; subjects with a dual nationality or a foreign nationality were classified as non-Italians. Statistically significant differences in seroprevalence were assessed according to sex, nationality, and age group, and also according to the vaccination status, number of doses, and time elapsed since the last dose. Statistical analyses were performed using the two-tailed chi-square test or Fisher’s exact test for comparing dichotomous variables. A Mantel-Haenszel trend test was applied to assess the presence of an increasing linear trend between serological status and age group (ordinal variable). Multivariate logistic regression analysis was performed including serological status as a dependent variable and sex, age (continuous variable), and nationality as independent variables.

The *p*-value 0.05 was considered statistically significant. All tests were performed using statistical analysis software JAMOVI 1.6.15.0 [16].

## 3. Results

The current study included 165 subjects aged between 1 and 18 years old, divided into four age groups, as shown in Table 1. The percentage of males and females were 53.3% and 46.7%, respectively. Most of the enrolled subjects were Italian citizens (83.1%), whereas the remaining part consisted of subjects with either dual (2.4%) or foreign citizenship (14.6%). All participants resided in 35 different districts of Florence and about 48% of them lived in the city of Florence.

### 3.1. Hepatitis A Antibodies Qualitative Measurement

The overall seroprevalence for anti-HAV antibodies was 9.1%. Table 2 shows the distribution of seroprevalence according to the sex and nationality. No significant difference was found between male and female subjects (*p* = 0.55) or between Italian and non-Italian subjects (*p* = 0.08) in the univariate analysis.

The seroprevalence in the different age groups is shown in Figure 1. The highest seroprevalence was observed in subjects aged 10–14 years old (18.0%, 9/50), the lowest in the 1–4 year-old ones (2.5%, 1/40). There was no linear growth trend in the number of positive subjects with increasing age (*p* = 0.27).

A statistically significant difference in seroprevalence was observed in the non-Italian group (higher seroprevalence) compared to individuals having an Italian nationality (lower seroprevalence) when the data were adjusted for age and sex in a multivariate logistic regression analysis. Sex and nationality, however, did not show a significant impact (Table 3).

### 3.2. Hepatitis A Notification, Vaccination Status, and Seroprevalence Assessment

According to the SIMI, no notifications for hepatitis A were recorded among the enrolled subjects. Furthermore, according to the questionnaire, 4/165 subjects did not recall if they had hepatitis A disease, while the rest of them responded negatively; no one answered affirmatively.

From the consultation of the regional vaccination registry, we observed that 7/165 subjects (4.2%) received at least one dose of anti-HAV vaccine. The 10–14 age group contained the highest number of vaccinated subjects (4/7), accounting for the 8% of the subjects in this age group. The percentage of vaccinated subjects in the male group was comparable to female (3.4% vs. 5.1%). Moreover, the percentage of vaccinated people among non-Italians was approximately twice than the Italian subjects (7.1% vs. 3.6%). However, this difference was not statistically significant (*p* = 0.40) (Table 4).

All vaccinated subjects tested positive for anti-HAV antibodies. Table 5 summarized the information on our study population by age group, serological result, and vaccination status.

According to the questionnaire, 6/165 subjects indicated that they had been vaccinated against hepatitis A, 7/165 subjects could not remember, and the majority (152/165) responded negatively. For 4/6 subjects who remembered being vaccinated against hepatitis A, data were confirmed by the regional vaccination registers. The remaining two individuals without confirmation of vaccination status, however, tested positive for anti-HAV antibodies (Table 6).

We also retrieved the number of vaccine doses each subject had previously received. Among the vaccinated subjects, 57.1% (4/7) received a single HAV vaccine dose, and 42.9% (3/7) received two doses. Almost all the vaccinated subjects, i.e., 85.7% (6/7) had received the last (or the only) dose up to more than 5 years before.

## 4. Discussion

The goal of the present study was to describe the susceptibility/immunity level to hepatitis A in a paediatric and adolescent population (1–18 years old) residing in the province of Florence (Italy). A second aim was to assess whether the seropositivity found derived from a natural infection or immunization. Our results were then compared to the national Italian average values and to a previous survey carried out in 2009 in the same geographical area. 

According to ISTAT (Italian National Institute of Statistics), the population between 1 and 18 years old residing in the province of Florence in 2018 accounted for 158,610 individuals, of which 25,930 (16%) held a foreign nationality [17,18]. Therefore, our sample population, accounting for 17% of foreign people, well represents the real demographic distribution of the Florentine paediatric and adolescent population.

Age-stratified seroprevalence allows indirect measurement of age-specific incidence rates of HAV infection and is considered the best way to evaluate the hepatitis A scenario in a country [3]. Seroprevalence is a useful indicator of population not at risk of developing the disease especially in a globalized world where migrants from endemic areas could be potentially high risk for spreading the disease to susceptible subjects in low endemicity areas [19,20,21].

The hepatitis A seroprevalence found in our study was equal to 9.1%, which is comparable to a low prevalence of hepatitis A. This is in line with WHO classification for Italy, and for many other European countries, as a low endemicity country for hepatitis A [22].

In the last decades, globalized nations showed epidemiological changes that require a full analysis. In particular, epidemiological transitions to lower endemicity levels lead to a reduced force of infection and shift in age of infection to later adulthood with the increase in the relative incidence of acute symptomatic hepatitis A. Understanding the mechanisms and the timing of epidemiological transitions in the EU and EEA regions might improve the assessment of the population currently at risk, the demand for adequate preventive measures, and also the need for tailored control programs to accelerate such transitions [23].⁠ Italy belongs to those countries that in the recent decades have undergone a drastic reduction of the HAV seroprevalence rate, thanks to improvement of socio-economic and sanitary conditions. This is particularly true for younger people, those who initially reflect the epidemiological transitions [24,25,26]. This shifting national phenomenon has also characterized the Tuscany region.

Stroffolini et al. evaluated the seroprevalence of the Italian population aged 3–19 years old between May 1987 and November 1989, and the overall prevalence rate of anti-HAV antibodies was 9.5%, in line with our results. Subsequently, data corresponding to five different geographical areas were evaluated, with higher rates found in southern Italy than in the northern part. A statistically significant linear growth trend in the seroprevalence was observed by Stroffolini (from 2.3% in 3–5 years to 16.3 in 17–19 years) [24]. This was not observed in our study in which a lower seroprevalence was found in the 15–18 year age group than in the 5–9 or the 10–14 year age groups. Moreover, in another study carried out at national level by Ansaldi et al., an increase in the anti-HAV antibody concentration was observed alongside age [27].

Moreover, the prevalence of anti-HAV antibodies on sera samples from subjects aged 1 to 20 years old was evaluated in Tuscany. Samples were obtained at three different times between 1992 and 2004 (1992, 1998, 2004) and showed an increase in seroprevalence in the 1–5 age group: from 2.7% in 1992 to 16.7% in 2004 [28]. This last value is higher than what we found in our study for the 1–4 year age group (2.5%), despite factors such as vaccination rate or immigration trend not varying. In the 15–20 year age group, Gentile et al. observed a reduction from 7.5% to 5.5%, in line with the results of our study. Catania et al. in 1996 [29] found a difference in seroprevalence between males and females in the younger age group of 0–12 years (15.3% male and 3.2% female). However, we did not find a statistically significant difference between the two sexes in any of the age groups considered. Indeed, the 2008 and 2017 outbreaks of Hepatis A virus highlighted its significant transmission burden among MSM (men who have sex with men) subjects in countries (like Italy) with a high number of young people susceptible to the virus. During these epidemics, the males were more affected than the females. The age of sexual debut in Italy is around 17 years [30], and our analysis showed that most subjects aged 15–18 years were not protected against the Hepatitis A virus and therefore remained susceptible to the disease. For these reasons, awareness campaigns about risk factors associated to hepatitis A infection, such as contaminated food or sexual intercourse, should be addressed to the younger population. Indeed, this part of the population is particularly at risk of contracting the infection because of travel to endemic areas, but also for the potential exposure to high-risk sexual practices.

In our analysis, we chose to classify the enrolled subjects into two groups by nationality, “Italian”/“non-Italian” because studies have demonstrated that subjects belonging to the group “visiting friends and relatives” (VFR), such as children of immigrants coming from countries with medium and high endemicity of hepatitis A, have a higher risk of contracting the disease [31], as they have a poor adherence to vaccination, despite the recommendations of the NIP 2017–2019. Scientific literature underlines the greater risk of children belonging to the VFR category of contracting HAV infection and the possibility to cause epidemic outbreaks in the country of residence on their return [32]. In Italy, the SEIEVA has observed a change in the epidemiological characteristics of the notified hepatitis A cases in the period 2018–2019 compared to those collected in the years before. In particular, many cases (included the secondary cases) were linked to travels in endemic areas (Morocco) and the estimated median aged was very low (10–12 years old). This suggested the possibility that early cases of HAV infection may have been associated with trips to Morocco and involved susceptible children visiting their country of origin. Then, once they have returned in Italy, they also started small outbreaks within the school setting [33]. The importance of a vaccination strategy targeting non-EU young travelers has been already supported by the scientific literature. For example, in the Netherlands, such interventions have been carried out on Turkish and Moroccan children starting from 1998, and a reduction in the number of hepatitis A cases, both imported and secondary, in various risk groups was reported [34]. 

In our sample there were 10 anti-HAV positive Italian children out of 137 subjects of Italian nationality (7.3%); whereas among 28 subjects of non-Italian nationality, 5 of them (17.9%) tested positive for anti-HAV antibodies. Our study therefore found a statistically significant difference in the seroprevalence for anti-HAV antibodies between children with Italian nationality and children with foreign nationality (*p* = 0.037).

Among the participants enrolled in our sample, only 4.2% (7/165) received the hepatitis A vaccination, as documented by the vaccination registry. This small number of vaccinated people is in line with the Tuscany regional guidelines regarding anti-HAV vaccination, recommended only for risk groups [7,35]. Despite the small number of vaccinated enrolled subjects, we found that vaccinated people among the non-Italian group were approximately twice that of the Italian ones (7.1% vs. 3.6%). This could reflect the current Tuscany Region recommendation for a free-of-charge vaccination to subjects traveling in endemic countries (which could be the case for children visiting their country of origin) [7].

The absence of a universal program of vaccination in Italy makes data unavailable for the vaccination coverage for all subjects aged 0–18 years on the national and regional territories. Instead, vaccination coverage is available for some single age cohorts. Data for the Tuscan vaccination coverage are available for cohorts aged 36 months, 16 years, and 18 years, and these could be compared with the findings of our analysis. In Tuscany, the vaccination coverage in 2017 was 0.90% for subjects aged 36 months, 2.60% for those aged 16 years, and 2.36% for subjects aged 18 years. Similar VCs were calculated for the same cohorts in 2018 [36]. In our sample, we observed that percentages of vaccinated subjects according to the age seem to in line with the regional data (0% in the 1–4 age group and 3.7% in the 15–18 age group). National vaccination coverages were higher as they were strongly influenced by VCs from Puglia Region.

Vaccine efficacy is supported by evidence in the literature, which reports a high, protective, and long-lasting immunogenicity for HAV vaccination. Almost 100% of subjects seroconvert 4 weeks after the first dose of HAV vaccine [37]. In addition, the effectiveness of vaccination is maintained even when the second dose is administered 27 months after the first one, confirming an excellent immunogenic profile [38]. Indeed, all the vaccinated subjects in our sample were positive to anti-HAV antibodies. Considering that 6/7 vaccinated subjects received the last dose at least 5 years before the enrolment in the study, our results highlighted the efficacy of the vaccine also in terms of duration, with detectable antibodies been maintained several years after the last dose was received. Moreover, 4/7 vaccinated subjects belonging to our sample received a single dose of HAV vaccine showing how even a single administration was effective in inducing detectable antibodies in the serum years later. The efficacy in terms of duration of even a single dose is the basis of the one-shot universal vaccination strategy adopted by some countries to contain costs [39,40].

The age group with the highest number of vaccinated subjects was the 10–14 year olds, and all vaccinated subjects in our sample received the last dose of HAV vaccine within 8 years of life. 

This study also demonstrates how the anamnestic memory of vaccination collected with the questionnaire at enrolment may be a useful tool to evaluate the state of immunity or susceptibility to a disease, integrating the data from the infectious disease registers and vaccination registry. Importantly, the analysis of questionnaires highlighted that two children, belonging to the group of subjects with protective antibody titers, stated that they did not contract the disease; whereas they recalled being vaccinated against HAV although this information was not recorded in the vaccination registry. Considering that no notification for hepatitis A was recorded for our study population, the presence of seropositive subjects among the unvaccinated group may be due to a natural immunization achieved through a subclinical infection and/or to underreporting of the surveillance or vaccination registries.

The prevalence study carried out by our research team has some limitations that must be taken into consideration.

Although the collected sample well represents the regional population in terms of composition by sex and nationality, it is numerically small, representing a sample of convenience. The test carried out at our laboratory is qualitative and highlights the presence or absence of an antibody titer higher than a reference cut-off. It is therefore not possible to obtain detailed information in terms of serum antibody concentration trends in the years following immunization. We could not retrieve information regarding adopted children with foreign nationality (or dual nationality), and it is challenging to assess whether they actually belong to the VFR category at risk of infection in case of travel to the countries of origin. 

## 5. Conclusions

The low percentage of young subjects positive for anti-HAV antibodies found in our study population suggests the low circulation of the HAV virus in the Florentine territory, showing a reduced rate of seroprevalence in the population 1–18 years old. Therefore, Tuscany remains a low hepatitis A endemic area, with the recommendation to immunize only high-risk groups.

The reduced HAV exposure at a young age may lead to a higher susceptibility in adulthood, which presents more frequently in a symptomatic form, sometimes severe. In addition, it is possible that HAV outbreaks occur in VFR children who expose themselves to the virus while traveling in countries with a medium/high prevalence of HAV. In fact, VFR travelers have a higher risk of being exposed to HAV as they are often unaware of pre-travel recommendations and may be in close contact with locals for long periods. Therefore, properly informing the VFR category could increase the adherence to the preventive measures for HAV.

Our study reported that subjects of foreign nationality or with dual nationality have a significantly higher prevalence of anti-HAV antibodies than children of Italian nationality. This supports the current recommendation by the NIP 2017–2019 and by the Tuscany Region health authority to vaccinate high risk groups, such as VFR [7].

## Figures and Tables

**Figure 1 vaccines-09-01194-f001:**
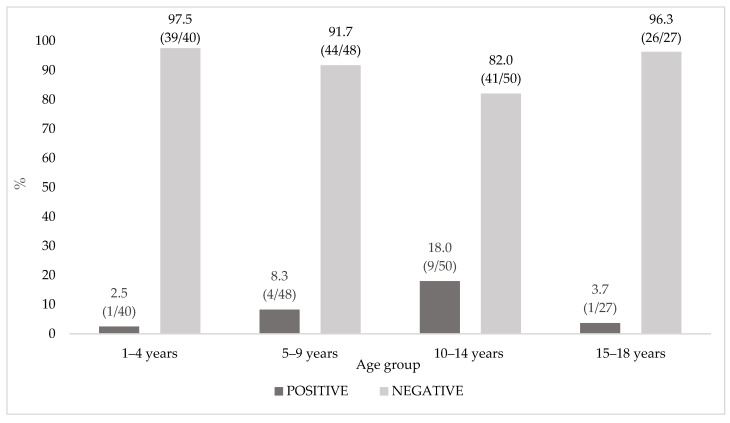
Seroprevalence distribution in the age groups. Note: Percentages and ratios (*n*/*N*) are shown above each bar.

**Table 1 vaccines-09-01194-t001:** Subjects distribution in the age groups of the study population according to sex and nationality.

Age Groups (years)	Nationality (*N*)		Sex (*N*)		Enrolled Subjects (*N*)
	Italian	Non-Italian	Male	Female	
1–4	27	13	19	21	40
5–9	41	7	27	21	48
10–14	43	7	27	23	50
15–18	26	1	14	13	27
TOTAL	137	28	87	78	165

**Table 2 vaccines-09-01194-t002:** Anti-HAV seroprevalence (*n*/*N*: number of subjects/total).

Anti-HAV Seroprevalence		
Group	Positive % (*n*/*N*)	Negative % (*n*/*N*)
Overall	9.1 (15/165)	90.9 (150/165)
Male	10.3 (9/87)	89.7 (78/87)
Female	7.7 (6/78)	92.3 (72/78)
Italian	7.3 (10/137)	92.7 (127/137)
Non-Italian	17.9 (5/28)	82.1 (23/28)

**Table 3 vaccines-09-01194-t003:** Multivariate logistic regression analysis.

Dependent Variable: Anti-HAV Positive					
		OR	SE	CI 95%	*p*-Value
**Sex**	Female	-	-	-	-
	Male	0.678	0.563	0.225–2.046	0.491
**Nationality**	Italian	-	-	-	-
	Non-Italian	0.264	0.641	0.075–0.926	0.037
**Age**	-	0.920	0.060	0.817–1.035	0.165

Note: OR: odds ratio; SE: standard error; CI: confidence interval; “-”: reference.

**Table 4 vaccines-09-01194-t004:** Vaccination status stratified by age group, sex, and nationality (*n*/*N*: number of subjects/total).

Variable	Group	Vaccinated % (*n*/*N*)	Unvaccinated % (*n*/*N*)
Age Group (years)	1–4	0.0 (0/40)	100.0 (40/40)
	5–9	4.2 (2/48)	95.8 (46/48)
	10–14	8.0 (4/50)	92.0 (46/50)
	15–18	3.7 (1/27)	96.3 (26/27)
Sex	Female	5.1 (4/78)	94.9 (74/78)
	Male	3.4 (3/87)	96.6 (84/87)
Nationality	Italian	3.6 (5/137)	96.4 (132/137)
	Non-Italian	7.1 (2/28)	92.9 (26/28)
	Total	4.2 (7/165)	95.8 (158/165)

**Table 5 vaccines-09-01194-t005:** Vaccination status of the enrolled subjects related to the serological result (*n*/*N*: number of subjects/total).

Anti-HAV Seroprevalence				
Group	Age Group (Years)	Positive % (*n*/*N*)	Negative % (*n*/*N*)	Total % (*n*/*N*)
Vaccinated	1–4	0.0 (0/0)	0.0 (0/0)	0.0 (0/7)
	5–9	100.0 (2/2)	0.0 (0/2)	28.6 (2/7)
	10–14	100.0 (4/4)	0.0 (0/4)	71.4 (5/7)
	15–18 Total	100.0 (1/1) 100.0 (7/7)	0.0 (0/1) 0.0 (0/7)	14.3 (1/7) 4.2 (7/165)
Unvaccinated	1–4	2.5 (1/40)	97.5 (39/40)	25.3 (40/158)
	5–9	4.3 (2/46)	95.7 (44/46)	29.1 (46/158)
	10–14	10.9 (5/46)	89.1 (41/46)	29.1 (46/158)
	15–18 Total	0.0 (0/26) 5.1 (8/158)	100.0 (26/26) 94.9 (150/158)	16.5 (26/158) 95.8 (158/165)

**Table 6 vaccines-09-01194-t006:** Concordance between anamnestic memory of vaccination and data from the regional vaccination registry.

Anti-HAV Antibodies			
Anti-HAV Vaccination According to Anamnestic Memory of Vaccination	Anti-HAV Vaccination According to Regional Vaccination Registry	Negative % (*n*/*N*)	Positive % (*n*/*N*)
No	No	96.7 (145/150)	3.3 (5/150)
	Yes	0.0 (0/2)	100.0 (2/2)
I don’t remember\I don’t know	No	83.3 (5/6)	6.7 (1/6)
	Yes	0.0 (0/1)	100.0 (1/1)
Yes	No	0.0 (0/2)	100.0 (2/2)
	Yes	0.0 (0/4)	100.0 (4/4)

## Data Availability

Data sharing not applicable. Data were collected and managed in aggregated form according to the European Union Regulation 2016/679 of the European Parliament and the Italian Legislative Decree 2018/101.

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
