# Peer review of "A Study of Hepatitis A Seroprevalence in a Paediatric and Adolescent Population of the Province of Florence (Italy) in the Period 2017–2018 Confirms Tuscany a Low Endemic Area"

_vaccines, 2021, doi:10.3390/vaccines9101194_

Round 1

Reviewer 1 Report

The manuscript by Zanella et al aims to study the prevalence to HAV in a cohort of pediatric population from the province of Florence in Italy.  The data of this study shows a seroprevalence of anti-HAV of 9.1% in the cohort, among them half have been vaccinated.  A statistically significant difference was observed between Italian and not-Italian groups.

Although this manuscript is well written and could constitute a good contribution to the field, I have nevertheless several specific comments:

  • The selection of patients during inclusion should be better described.  Were all patients who had blood sampling at Meyer Children's Hospital during the time period of December 2017 to April 2018 included in the study? if not, what were the inclusion criteria? this information is very important to exclude that there was no selection bias
  • The authors should further precise how patients were stratified according to nationality. Are considered not-Italian all foreign people, regardless of the time spend in Italy? What about the children born in Italy?
  • Table 1 should be completed with information on gender, nationality in each age category.
  • In table 2, it is not necessary to give the data for both the positives and the negatives, since these results are complementary and therefore giving both is redundant. However, it would be interesting for the reader to have the p-values in this table
  • Since the authors discuss extensively the increased risk of contracting HAV in the non-Italian population, it would be important to mention whether there is a difference in vaccination status between the two populations. The only information provided to us is that the seroprevalence is higher in the not-Italian population, but are they more vaccinated? This could be a possibility as the guidelines recommend a free of charge vaccination for children traveling in endemic countries (which could be the case for children visiting their country of origin).
  • Did the survey include questions about possible travel to countries considered endemic?
  • I don't quite understand how the calculations were done in tables 4 and 5. Indeed for the column "total %, I think it would be more relevant to give the percentages of vaccinated vs non vaccinated patients in each age group, e.g. in the 5-9 years old, there are two vaccinated patients among the 48 patients of this age group (2/48 --> 4.1%).  In addition, in this same column, there is an error for the 10-14 age group, since there are 4 patients and not 5 who are vaccinated.  Again, it is not necessary to give the data for both the positive and the negative patients.
  • I also noticed an error in the numbers in table 5: according to the text, there are 152 patients who answered negatively. However, when we add up the data of the vaccination records given in the table, we have 148 "no" and 2 "yes" which makes a total of 150 and not 152. Could you please correct this? 
  • Did HAV seropositive patients who were not vaccinated (table 5) report a natural infection or were mentioned in the infectious disease surveillance records ? Could the author give more information on this group of patients ?

Minor comments :

  • Could the authors provide the information concerning the vaccination rate in Italy in 2017-2018 potentially available in the national registries. As well as in the other province of Tuscany? This data would be interesting to compare with those of the current study.
  • Tables 3 to 6 are misnumbered
  • How many patients fell into the "gray area" category?

Reviewer 2 Report

This is an interesting article about the Hepatitis A seroprevalence in a pediatric and adolescent population of Italy. I would like to address a small number of suggestions to you.

Page 3 line 150-151

This sentence "For the purposes of the analysis, all subjects holding an Italian nationality were classified as Italians; subjects with a dual nationality or a foreign nationality were classified as not-Italians." may added before excluding criteria

Page 6 line 203

You write that in the Table 4 are summarized the information on your study population by age group, serological and vaccination status.

Some information is represented also in table 2 (Vaccination status of the enrolled subjects). For only 7 vaccinated persons, two tables are unnecessary. Moreover, data about the doses are presented in the text, lines 218-221.

Is travel histories were taken from your cohort?

Please correct references according to the ACS style guide.

References should be described as follows

Journal Articles:

  1. Author 1, A.B.; Author 2, C.D. Title of the article. Abbreviated Journal Name Year, Volume, page range.
